# Past-Year Blunt Smoking among Youth: Differences by LGBT and Non-LGBT Identity

**DOI:** 10.3390/ijerph20075304

**Published:** 2023-03-29

**Authors:** Robert Andrew Yockey, Tracey E. Barnett

**Affiliations:** School of Public Health, University of North Texas Health Science Center, Fort Worth, TX 76107, USA

**Keywords:** blunts, LGBT, youth

## Abstract

Blunt use (co-use of tobacco and marijuana) is a growing phenomenon among youth and disproportionately affects minority populations. LGBT+ populations are significantly more likely to use marijuana and tobacco, but this relationship has yet to be examined among LGBT+ adolescents. This analysis aimed to investigate past-year blunt use among a national sample of youth and delineate the differences between non-LGBT and LGBT+ youth. We used Wave 2 of the Population and Tobacco Health (PATH) study. We analyzed data from 7518 youth, comparing past-year blunt use between LGBT+ and non-LGBT youth, controlling for biological sex, race, and age using weighted logistic regression models. Greater than 1 in 10 youth (10.6%) reported using blunts in the past year. More than one in five (21.6%) LGBT+ youth reported using blunts in the past year. There were no significant differences between boys and girls. Older youth (17 years old) were more likely to use blunts in the past year (aPR: 3.04, 95% CI 2.48, 3.79) than younger youth. Compared with non-LGBT youth, LGBT+ youth were 2.17 times (95% CI 1.86, 2.54) more likely to report using blunts in the past year. Blunt use and its respective impact on health outcomes among developing youth are of concern to public health. These findings demonstrate that certain subgroups of youth are more at risk for use and emphasize the need for tailored interventions to mitigate initiation and current use, given that one of the goals of the Healthy People 2030 initiative is to “Improve the health, safety, and well-being of lesbian, gay, bisexual, and transgender (LGBT) individuals”.

## 1. Introduction

The availability and widespread legalization of cannabis in the United States has culminated in several different variations in use (e.g., blunts and edibles) among different populations, including youth and young adults. For example, blunts, which are hollowed-out cigars filled with cannabis, are frequently used among youth and young adults, with previous research estimating that between 12% and 19% reported blunt use in the past month [1,2]. Although perceived to be a greater risk to one’s health and perceived to have greater social consequences such as getting into trouble (compared with other cannabis products such as edibles [3]), a recent article reported a substantial increase (14.9%) in current blunt use among Florida youth between 2015 and 2018 [4], highlighting the possible popularity, initiation, and appeal among this population [5].

The co-use of tobacco and marijuana has several detrimental effects, including exposure to carcinogens, the elevated risk of dependence, increased exposure to several different strains of bacteria [6], and pulmonary abnormalities (e.g., wheezing, shortness of breath) [7]. One study of nicotine wrappers used for blunts found that users were exposed to high levels of nicotine, cotinine, and other nitrosamines, which are comparable to high levels of secondhand smoke exposure [8]. Another study of young adults specifically found that those who used blunts had a higher progression to using cigars, thereby possibly exposing them to a higher dependence on tobacco [9]. Further, that same study found that ever using a cigar was associated with a 14-fold increase in the odds of progressing to 30-day blunt use [9].

Moreover, both marijuana and cigarettes are reported to have higher rates of use among young adults (18–25) who identify as LGBT+ vs. those who do not [10]. Sexual minority young adults’ past-year marijuana use was 46.2%, compared with that of sexual majority adults (31.0%); and past-month cigarette use among sexual minority young adults was 39.7% vs. 25.5% reported among sexual majority young adults [10].

Sociodemographic characteristics (e.g., sex, race, etc.) have also been found to be strong predictors of blunt use. One study of US youth aged 12–17 years old reported Hispanic and African American youth had higher odds of lifetime blunt use and current daily blunt use compared with White youth [11]. Another study demonstrated that African American young adults were more likely to initiate blunt use at an earlier age, reported more days of blunt smoking, and smoked more blunts than their other ethnic counterparts [12]. To date, there have been no studies that have examined LGBT+ youth and blunt use, which may be problematic given the higher reported use rates of marijuana and tobacco products among LGBT+ populations, compared with non-LGBT+ populations [10].

Therefore, the present analysis examines the effects of sexual minority status on past-year blunt use among a large sample of youth in the United States and estimates the associated demographic characteristics. We hypothesize that compared with non-LGBT youth, LGBT+ youth would report higher past-year blunt use. We also hypothesize that compared with females, males would report higher blunt use, based on the prior literature, and that there would be racial/ethnic differences in terms of past-year use [11].

## 2. Methods

The present study was a secondary analysis of the PATH (Population and Tobacco Health) survey, Wave 2. Briefly, the PATH survey is a nationally representative, longitudinal cohort study of youth, tobacco use patterns, and its health effects on people aged 9 or older in the United States and the District of Columbia. The PATH uses a complex sampling survey design to ensure an adequate probability of being selected. Specifically, a four-stage, stratified probability sample design was employed involving the selection of sampling units and mailing addresses to ensure equal probabilities of response by participants. Once selected, households within the sampling units and mailing addresses were selected for participation. The PATH uses computer-assisted personal interviewing (CAPI) and audio-computer-assisted self-interviewing (ACASI) methods to ensure the privacy of responses and to ensure participant confidentiality. Additional details about the PATH are explained elsewhere [13]. We chose to use Wave 2 because it was the first wave that asked about transgender identity status [13]. The weighted youth response rate for Wave 2 was 87.3%. The sample was further delimited to youth aged 14–17 since 12–13-year-olds were not queried about sexual orientation.

## 3. Measures

### 3.1. Dependent Variable—Past-Year Blunt Use

The outcome was past-year blunt use, which was ascertained by the following question: “Sometimes people take tobacco out of a traditional cigar, cigarillo or filtered cigar and replace it with marijuana. This is sometimes called a “blunt”. In the past 12 months, have you smoked part or all of a traditional cigar, cigarillo, or filtered cigar with marijuana in it?” The answers were “Yes”, “No”, “Don’t Know”, or “Refused”. For the purposes of this study, we removed “Don’t Know (*n* = 9) and “Refused” (*n* = 11).

#### Sexual Identity/Transgender Status

To assess sexual identity, the following question was asked from youth (14–17): “Do you consider yourself to be...”. The response options were “Straight”, “Lesbian or Gay”, “Bisexual”, or “Something else”. To assess transgender identity, the following question was asked from youth: “Some people describe themselves as transgender when they experience a different gender identity from their sex at birth. For example, a person born into a male body, but who feels female or lives as a woman would be transgender. Do you consider yourself to be transgender?” The response options were “Yes” or “No”. For the purposes of this study, we combined sexual identity and transgender status to create a composite LGBT+ youth variable, with non-LGBT+ youth serving as the reference category.

### 3.2. Demographics

Participants’ biological sex (Male, Female), age (14, 15, 16, and 17 years old), and race/ethnicity (White, African American, Asian, and Mixed Race/Other) were used as covariates. Mixed Race/Other was a combination of individuals provided by the PATH (including Hispanic, Vietnamese, Filipino, Other Asian, Native Hawaiian, and Other Race). 

### 3.3. Analysis Plan

Descriptive statistics with 95% confidence intervals were estimated to capture participant demographics and past-year blunt use. We opted to perform a complete case analysis, since <5% of data were missing on variables of interest [14]. Bivariate comparisons were performed with a Rao–Scott correction. We estimated multivariable generalized linear models using Poisson distribution and log link to estimate the adjusted prevalence ratios (aPRs). All estimates incorporated the sampling weights provided by the PATH for representative estimates. Variance estimates were estimated with the balance repeated replication method with Fay’s adjustment set to 0.3 [15,16]. All analyses were conducted in Stata 17.0. A *p* < 0.05 was considered statistically significant. A University Institutional Review Board approved this study.

## 4. Results

### 4.1. Demographics and Bivariate Comparisons

The sample comprised 7518 youth, with more boys than girls (51.5% vs. 48.5%, respectively). An estimated 10.6% of youth aged 14–17 (*n* = 800) reported past-year blunt use. There were significant differences based on race/ethnicity (*p* < 0.0001), age (*p* < 0.0001), and LGBT+ status (*p* < 0.0001). No significant differences were found based on biological sex. Notably, more than one in five (21.6%; (95% CI: 18.8, 24.8)) LGBT+ youth reported past-year blunt use. See Table 1 for estimates. 

### 4.2. Final Multivariate Model

Compared with White youth, African American youth were 1.26 (95% CI: 1.03, 1.55) times more likely to report past-year blunt use. Further, youth who identified as Mixed Race/Other were 1.38 (95% CI: 1.44, 1.66) times more likely to use blunts in the past year. Asian youth were less likely to use blunts in the past year (aPR: 0.28, 95% CI: 0.11, 0.73). There were no significant differences between males and females regarding past-year blunt use (aPR: 0.97, 95% CI: 0.85, 1.11) (see Table 2). As age increased, the likelihood of reporting past-year blunt use increased, with 17-year-olds 3.04 times more likely (95% CI: 2.48, 3.79) to report blunt use compared with 14-year-olds. Youth who identified as LGBT+ were 2.17 times (95% CI 1.86, 2.54) more likely to report using blunts in the past year compared with non-LGBT+ youth.

## 5. Discussion

### 5.1. Principal Findings

Greater than 1 in 10 (10.6%) of youth aged 14–17 reported past-year blunt smoking. Demographic analyses revealed that more than one in five (21.6%) LGBT+ youth aged 14–17 reported past-year blunt use, and multivariate results revealed that LGBT+ were twice as likely to report use compared with non-LGBT+ youth. Multivariate results also demonstrated that African American and multiracial youth were more likely to report blunt use than White youth, and older youth (aged 17) were more likely than younger youth (aged 14). Blunt use is a commonly used combined marijuana and tobacco product among youth, with particularly vulnerable youth demonstrating higher use, which may lead to exposure to harmful bacteria [6] and increased risk for worsening pulmonary functioning [17] and adverse cardiovascular outcomes [18].

### 5.2. Findings in Context

We found an increase in the prevalence of blunt use and higher risk as youth aged, with an increase in odds as age increased by year. Similarly, one study found that among young adults, risk continued to increase as respondents aged, with higher use among older young adults compared with younger young adults [19]. Given the popularity of blunt use on several social media sites [18] and in youth culture [12,20], as well as the fact that most drug initiation occurs in adolescence [21], and that blunts are seen as less harmful than other tobacco products [22], focusing on the harms of blunt use (e.g., dependence, pulmonary problems, etc.) at an earlier age may deter use. 

Compared with White youth, the findings estimated racial/ethnic differences, such that African American and Mixed Race/Other youth were at higher risk, corroborating previous research [11], while Asian youth were at lower risk for the use of blunts in the past year. History reveals that blunt use and blunt culture were centered among Black males in the inner-city [23] and were a prime target for big tobacco companies to initiate messaging surrounding the positive aspects of tobacco [24]. Further, cigar products used for blunts have regularly been marketed to youth and minorities [5], which may expand the initiation and appeal of blunt products to developing youth. Prior studies have revealed that African American youth are most at risk for blunt use [11], and thus should be a prioritized population for intervention, considering they view blunts as safe alternatives to other tobacco products [25]. Interventions are needed to address harm reduction approaches toward marijuana and tobacco products among minority populations [26].

Finally, the main research question demonstrated that youth who identified as LGBT+ were more likely to report blunt use in the past year, compared with their non-LGBT+ counterparts, even when accounting for other demographic correlates. This may be in large part because LGBT+ youth are more likely to initiate the use of marijuana and associated products at an earlier age, compared with their non-LGBT+ counterparts [27]. Likewise, LGBT+ youth may engage in higher blunt use because of their social environment. One study of adolescents [28] examined the social contexts of blunt smoking and found that group smoking was an integral part of blunt use initiation, as this was a way to increase social bonding. This may extend to LGBT+ youth, a group that reports low levels of social support and high levels of loneliness, as a way to seek inclusion and social bonding.

Moreover, minority stress theory (MST) posits that a culmination of stress will ensue due to feeling ostracized from the general population [29], and this may lead to engaging in behaviors (e.g., blunt smoking) to reduce stress. Interventions are needed to reduce blunt use among LGBT+ youth, given that one of the goals of the Healthy People 2030 initiative is to “Improve the health, safety, and well-being of lesbian, gay, bisexual, and transgender (LGBT) individuals” [30].

## 6. Limitations

This study is not without limitations. One limitation is the limited ages of 14–17, which does not represent the full range of adolescence (12–17) often presented in research. The result is a smaller comparison group that may not be generalizable to youth or comparable with others’ work. However, given the large prevalence as well as the higher use among older youth, the findings of this study are relevant. Additionally, the primary research question regarding blunt use among LGBT+ youth is the reason for the smaller dataset to include that valuable information. Another potential limitation is the categorization of participants as either LGBT+ or non-LGBT+ based on a single question. While this approach is common in survey research, it may not fully capture the complexity of sexual and gender identities. Additionally, this study did not assess other potentially important variables such as gender expression, which may impact the relationship between LGBT+ status and blunt use. Future research is also needed on a larger LGBT+ population to capture additional differences. Further limitations include the fact that youth may have confusion regarding the difference between blunts and cigarillos [31], which may lead to under-/over-reporting of use. This study may be limited by the specific questions asked in the PATH survey, which may not fully capture the complexity of youth tobacco use patterns and the impact on health outcomes. Additionally, it may be limited by the self-report nature of the data, which could be subject to recall bias and social desirability bias. Lastly, data were cross-sectional; therefore, casual implications are limited. 

One substantial strength of the study is that it includes a large, nationally representative sample of youth including information about LGBT status. Future studies should include sexual identities to properly capture the heterogeneity of LGBT+ youth.

## 7. Conclusions

Greater than 1 in 10 youth reported the use of blunts in the past year, with LGBT+ and racial minorities youth at the highest risk for reporting use. Future studies are warranted on disentangling the relationship between sexual identity status and blunt use among youth. Specifically, additional analyses (e.g., longitudinal) are warranted to identify the specific risk factors associated with blunt use through a theoretical lens to guide educational initiatives. The findings of this study also have implications for clinical interventions and future research initiatives. 

## Figures and Tables

**Table 1 ijerph-20-05304-t001:** Demographics and Bivariate Comparisons.

	Univariable	Bivariable Comparisons	
	Full Sample (*n* = 7518) Weighted % (95% CI)	No Past-Year Blunt Use (*n* = 6718) Weighted % (95% CI)	Past-Year Blunt Use (*n* = 800) Weighted % (95% CI)
Biological Sex			
Male	51.5 (51.1, 51.9)	89.8 (88.6, 90.4)	10.2 (9.15, 11.4)
Female	48.5 (48.1, 48.8)	89.0 (87.9, 90.0)	11.0 (10.0, 12.1)
Age			
14 Years Old	25.4 (24.8, 25.9)	94.4 (93.2, 95.3)	5.64 (4.67, 6.78) ***
15 Years Old	25.6 (25.0, 26.2)	91.4 (89.8, 92.7)	8.61 (7.28, 10.2)
16 Years Old	25.4 (24.6, 26.2)	88.3 (86.6, 89.7)	11.7 (10.3, 13.4)
17 Years Old	23.7 (23.1, 24.3)	83.1 (81.1, 85.0)	16.9 (15.0, 18.9)
Race/Ethnicity			
White	70.2 (69.5, 70.8)	89.8 (88.9, 90.8)	10.2 (9.19, 11.2) ***
Black/African American	15.5 (15.2, 15.8)	87.2 (84.9, 89.3)	12.8 (10.7, 15.1)
Asian	4.68 (4.50, 4.87)	97.3 (93.4, 98.9)	2.74 (1.10, 6.63)
Mixed/Other	9.67 (9.14, 10.2)	85.8 (83.2, 88.0)	14.2 (12.0, 16.8)
LGBT+			
Non-LGBT+	90.6 (89.8, 91.3)	90.5 (89.7, 91.3)	9.46 (8.70, 10.3) ***
LGBT+	9.40 (8.66, 10.2)	78.4 (75.2, 81.2)	21.6 (18.8, 24.8)

*** *p* < 0.001.

**Table 2 ijerph-20-05304-t002:** Multivariate regression results.

Variable	aPR	95% CI
Biological Sex		
Male	1.00	Ref.
Female	0.97	[0.85, 1.11]
Race		
White	1.00	Ref.
Black/African American	1.26	[1.03, 1.55] *
Asian	0.28	[0.11, 0.73] **
Other Race, including multiracial	1.38	[1.14, 1.66] ***
Age		
14 years old	1.00	Ref.
15 years old	1.55	[1.21, 1.97] ***
16 years old	2.09	[1.66, 2.64] ***
17 years old	3.04	[2.48, 3.79] ***
Sexual Orientation		
Heterosexual	1.00	Ref.
LGBT+	2.17	[1.86, 2.54] ***

*** *p* < 0.001; ** *p* < 0.01, * *p* < 0.05.

## Data Availability

Publicly available datasets were analyzed in this study. This data can be found here: https://www.icpsr.umich.edu/web/NAHDAP/studies/36231 (accessed on 3 November 2022).

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
