# Peer review of "Past-Year Blunt Smoking among Youth: Differences by LGBT and Non-LGBT Identity"

_ijerph, 2023, doi:10.3390/ijerph20075304_

Round 1

Reviewer 1 Report

The study provides important insights into the association between LGBT status and past-year blunt use among youth in the United States. The findings suggest a need for targeted prevention and intervention efforts for LGBT youth who may be at increased risk for substance use.

The study is rather simple, and answers only one research question. Why a more complex researh is not conducted, that would answer why? Is this a part of a larger study?

The authors do not provide information about the goodness-of-fit of their models, which could be assessed using measures such as the deviance or Pearson chi-square statistics.

The authors should provide more information about the limitations of using a secondary data source. For example, the study may be limited by the specific questions asked in the PATH survey, which may not fully capture the complexity of youth tobacco use patterns and the impact on health outcomes. Additionally, the study may be limited by the self-report nature of the data, which could be subject to recall bias and social desirability bias.

One potential limitation is the reliance on self-reported data, which may be subject to recall bias and social desirability bias. Additionally, the study is limited to a specific age range (14-17 years old) and may not be generalizable to other age groups. The study is also limited to a specific population (youth in the United States) and may not be generalizable to other populations.

Another potential limitation is the categorization of participants as either LGBT or non-LGBT based on a single question. While this approach is common in survey research, it may not fully capture the complexity of sexual and gender identities. Additionally, the study did not assess other potentially important variables such as gender expression, which may impact the relationship between LGBT status and blunt use.

There is a lack of future studies section, which this study requires. 

Author Response

Response to Reviewer 1

“The study is rather simple, and answers only one research question. Why a more complex research is not conducted, that would answer why? Is this a part of a larger study?”

Author’s Revisions:. We will continue this research in the future, but this is one of the first studies examining these relationships. We sought to assess if there was a relationship that warranted further study. To further investigate these relationships, future studies should examine longitudinal components utilizing multiple waves of data.

“The authors do not provide information about the goodness-of-fit of their models, which could be assessed using measures such as the deviance or Pearson chi-square statistics.”

Author’s Revisions: Using the survey commands in Stata, this information is not available. Ways to further develop this model will be future work. We also did not use chi-square statistics, given that this statistic is very sensitive to large samples and would not be an adequate measure of model fit.

“The authors should provide more information about the limitations of using a secondary data source. For example, the study may be limited by the specific questions asked in the PATH survey, which may not fully capture the complexity of youth tobacco use patterns and the impact on health outcomes. Additionally, the study may be limited by the self-report nature of the data, which could be subject to recall bias and social desirability bias.”

 “One potential limitation is the reliance on self-reported data, which may be subject to recall bias and social desirability bias. Additionally, the study is limited to a specific age range (14-17 years old) and may not be generalizable to other age groups. The study is also limited to a specific population (youth in the United States) and may not be generalizable to other populations.”

“Another potential limitation is the categorization of participants as either LGBT or non-LGBT based on a single question. While this approach is common in survey research, it may not fully capture the complexity of sexual and gender identities. Additionally, the study did not assess other potentially important variables such as gender expression, which may impact the relationship between LGBT status and blunt use.”

Author’s Revisions: Thank you for these comments. Taken together, these three comments resulted in a more fully developed limitations section. The limitations have been added to the paper and the section now reads:

The study is not without limitations. One limitation is the limited ages of 14-17, which does not represent the full range of adolescence (12-17) often presented in research. The result is a smaller comparison group that may not be generalizable to youth or comparable with others’ work. However, given the large prevalence as well as the higher use among older youth, the findings are relevant. Also, the primary research question regarding blunt use among LGBT+ youth is the reason for the smaller dataset to include that valuable information. Another potential limitation is the categorization of participants as either LGBT + or non-LGBT + based on a single question. While this approach is common in survey research, it may not fully capture the complexity of sexual and gender identities. Additionally, the study did not assess other potentially important variables such as gender expression, which may impact the relationship between LGBT + status and blunt use. Future research is also needed on a larger LGBT+ population to capture additional significant differences. Further limitations include youth may have confusion regarding the difference between blunts and cigarillos [28], which may lead to under/overreporting of use. The study may be limited by the specific questions asked in the PATH survey, which may not fully capture the complexity of youth tobacco use patterns and the impact on health outcomes. Additionally, the study may be limited by the self-report nature of the data, which could be subject to recall bias and social desirability bias. Lastly, data were cross-sectional; therefore, casual implications are limited.

“There is a lack of future studies section, which this study requires.” 

Author’s Revisions: We thank the reviewer for this comment. We have added a “Future Studies” section in the discussion.

Reviewer 2 Report

This is a very interesting study which is well written, with the potential to make a strong contribution to scholarship. The study is justified, it has a clear research question and the use of existing data from a larger study is appropriate.

What the study is lacking at the moment is a more thorough explanation of the finding that youth who identify themselves with the LGBT community are more likely to use blunts. The short explanation in the Discussion section is underdeveloped.

Also, LGBT, as an abbreviation, has been expanded in the recent years to include more gender and sexual identities; the most extended version of it should show in the paper.

Font size or type looks different in lines 136,137 and 138.

Author Response

“Font size or type looks different in lines 136,137 and 138.”

Author’s Revisions: We thank the reviewer for this comment. This has been fixed to ensure consistency.

“Also, LGBT, as an abbreviation, has been expanded in the recent years to include more gender and sexual identities; the most extended version of it should show in the paper.”

Author’s Revisions: We thank the reviewer for this comment. This has been fixed to ensure consistency and now reads LGBT+ (https://www.rethink.org/advice-and-information/living-with-mental-illness/wellbeing-physical-health/lgbtplus-mental-health/).

“What the study is lacking at the moment is a more thorough explanation of the finding that youth who identify themselves with the LGBT community are more likely to use blunts. The short explanation in the Discussion section is underdeveloped.”

Author’s Revisions: We thank the reviewer for this comment. We have added additional information to the discussion, including two more sources, to better highlight the relationship between LGBT+ adolescents and blunt use.

Reviewer 3 Report

 The manuscript entitled “Past year blunt smoking among youth: Differences by LGBT  and non-LGBT status” contains an interesting study in developmental psychology in the context of public health. The study is a valuable contribution to the extant literature. However, in my modest opinion, a few issues need to be addressed before publication is advised.

 The abstract needs to describe more accurately key terms. For instance, what is “blunt smoking”? What is the rationale of the study? Namely, why is the distinction between non-LGBT youth and LGBT youth relevant? Are these populations differentially at risk? The authors put forth a rationale in the introduction, but fail to mention it here.

 The introduction does not contain a sufficient review of the extant literature. The authors mention that “[t]he co-use of tobacco and marijuana has several detrimental effects”. Then, they offer a list without providing an accurate description of the studies and the evidence upon which each of the purported detrimental effects rests. They also fail to put the phenomenon of interest in the broader context provided by other substances and their use by adolescents.  If socio-demographic factors differentiate users, is there an account for such differences? Are edibles safer? How do the effects of edibles compare with those of blunt smoking?

 Before the method section, a paragraph devoted to the hypotheses and their respective rationale must be introduced. Because the authors also look at ethnicity, age, and gender in their data analyses, a broader set of hypotheses can be formulated.

 In the method section, the survey methodology of PATH (Population and Tobacco Health) survey, Wave 2 must be described more accurately.

 In the data analysis, the sample of non-LGBT youth is substantially larger than the sample of LGBT youth. Can the differences in sample size have affected the results of the study?

In the discussion section, a broader review of the extant literature on the detrimental effects of blunk smoking is to be introduced. Most importantly, the authors need to consider different types of impacts,  including the physiological, psychological (cognitive and socio-emotional functioning), and societal spheres.

Author Response

“The abstract needs to describe more accurately key terms. For instance, what is “blunt smoking”? What is the rationale of the study? Namely, why is the distinction between non-LGBT youth and LGBT youth relevant? Are these populations differentially at risk? The authors put forth a rationale in the introduction, but fail to mention it here.”

Author’s Revisions: We thank the reviewer for this comment. We have added a more detailed rationale for the purpose of the study.

“The introduction does not contain a sufficient review of the extant literature. The authors mention that “[t]he co-use of tobacco and marijuana has several detrimental effects”. Then, they offer a list without providing an accurate description of the studies and the evidence upon which each of the purported detrimental effects rests. They also fail to put the phenomenon of interest in the broader context provided by other substances and their use by adolescents.  If socio-demographic factors differentiate users, is there an account for such differences? Are edibles safer? How do the effects of edibles compare with those of blunt smoking?”

Author’s Revisions: We thank the reviewer for this comment. We have added additional detail of the studies. We have also included language regarding the perceived risk/harm of edibles versus blunts.

“Before the method section, a paragraph devoted to the hypotheses and their respective rationale must be introduced. Because the authors also look at ethnicity, age, and gender in their data analyses, a broader set of hypotheses can be formulated.”

Author’s Revisions: We thank the reviewer for this comment. We have added a paragraph devoted to the hypotheses and rationale. The section now includes:

Therefore, the present analysis examines the effects of sexual minority status on past year blunt use among a large sample of youth in the United States and estimates associated demographic characteristics. We hypothesize that compared to non-LGBT youth, LGBT+ youth would report higher past-year blunt use. We also hypothesize that compared to females, males would report higher blunt use, based on prior literature, and that there would be racial/ethnic differences in terms of past-year use [10].

“In the method section, the survey methodology of PATH (Population and Tobacco Health) survey, Wave 2 must be described more accurately.” 

Author’s Revisions: We thank the reviewer for this comment. We have added several details to the PATH survey design in the methods section. The section now reads:

The PATH uses a complex sampling survey design to ensure adequate probability of being selected. Specifically, a four-stage, stratified probability sample design involving the selection of sampling units and mailing addresses was employed to ensure equal probabilities of response by participants. Once selected, households withing the sampling units and mailing addresses were selected for participation. The PATH uses computer-assisted personal interviewing (CAPI) and audio-computer-assisted self-interviewing (ACASI) methods to ensure privacy of responses and to ensure participant confidentiality. Additional details about PATH are explained elsewhere. [12]

“In the data analysis, the sample of non-LGBT youth is substantially larger than the sample of LGBT youth. Can the differences in sample size have affected the results of the study?”

Author’s Revisions: We thank the reviewer for this comment. We have added this as a possible limitation to the study.

“In the discussion section, a broader review of the extant literature on the detrimental effects of blunt smoking is to be introduced. Most importantly, the authors need to consider different types of impacts, including the physiological, psychological (cognitive and socio-emotional functioning), and societal spheres.”

Author’s Revisions: We thank the reviewer for this comment. We have added language regarding these impacts on blunt use as well as additional citations.